# Investigating the Improvement of Autonomous Vehicle Performance through the Integration of Multi-Sensor Dynamic Mapping Techniques

**DOI:** 10.3390/s23052369

**Published:** 2023-02-21

**Authors:** Hyoduck Seo, Kyesan Lee, Kyujin Lee

**Affiliations:** 1College of Electronics & Information, Kyunghee University, 1732 Deogyeong-daero, Giheung-gu, Yongin-si 17104, Gyeonggi-do, Republic of Korea; 2Department of Electronic Engineering, Semyung University, 65 Semyung-ro, Jecheon-si 27136, Chungcheongbuk-do, Republic of Korea

**Keywords:** mobile mapping system (MMS), autonomous driving, dynamic high-definition map

## Abstract

The emergence of autonomous vehicles marks a shift in mobility. Conventional vehicles have been designed to prioritize the safety of drivers and passengers and increase fuel efficiency, while autonomous vehicles are developing as convergence technologies with a focus on more than just transportation. With the potential for autonomous vehicles to serve as an office or leisure space, the accuracy and stability of their driving technology is of utmost importance. However, commercializing autonomous vehicles has been challenging due to the limitations of current technology. This paper proposes a method to build a precision map for multi-sensor-based autonomous driving to improve the accuracy and stability of autonomous vehicle technology. The proposed method leverages dynamic high-definition maps to enhance the recognition rates and autonomous driving path recognition of objects in the vicinity of the vehicle, utilizing multiple sensors such as cameras, LIDAR, and RADAR. The goal is to improve the accuracy and stability of autonomous driving technology.

## 1. Introduction

Autonomous driving, also referred to as self-driving or driverless technology, is a rapidly evolving field that seeks to enable vehicles to operate without human intervention. The development of autonomous driving is driven by advancements in multiple technological domains, including sensors, machine learning, artificial intelligence, and data processing and connectivity. The primary objective of autonomous driving is to enhance road safety and increase transportation efficiency by reducing the number of accidents caused by human error and by optimizing traffic flow and minimizing congestion [1].

Autonomous driving technology is a highly interdisciplinary field that draws upon a range of disciplines including engineering, computer science, psychology, and robotics. The private sector and academic institutions are investing a significant amount of research and development into the development of autonomous driving technology [1]. Despite the rapid growth in the deployment of semi-autonomous vehicles on the roads, fully autonomous vehicles are still facing numerous technical and regulatory challenges, including ensuring system reliability and security, addressing privacy concerns, and establishing clear regulations and standards [2].

Dynamic maps, also referred to as high-precision or HD maps, are a critical component of autonomous driving technology. These maps provide real-time information about the road environment, including lane markings, road geometry, traffic signs, and obstacles. The information provided by dynamic maps is crucial for enabling autonomous vehicles to make informed driving decisions and navigate roads safely and efficiently [3].

Dynamic maps differ from traditional maps in that they are constantly updated in real-time based on data gathered by various sensors onboard the autonomous vehicle. The information gathered by these sensors, including LIDAR, RADAR, and cameras, is used to create a three-dimensional representation of the road environment. LIDAR provides detailed 3D information about the road environment, while RADAR is used to detect obstacles and measure their speed and direction. Cameras offer high-resolution color and texture information. The use of a combination of different sensors, such as RADAR, LIDAR, and cameras, is necessary for the creation of dynamic maps [4].

Dynamic maps are a fundamental component of autonomous vehicle technology, as they provide real-time representation of the driving environment. This information is essential for enabling autonomous vehicles to make informed decisions and navigate roads safely and effectively. Dynamic maps are constantly updated with the latest information, including details about road conditions, traffic patterns, road layout, traffic lights, road signs, and other key features [5].

The integration of dynamic maps into autonomous vehicles offers several key benefits. Firstly, by providing real-time information about road conditions, traffic, and other factors, dynamic maps enable autonomous vehicles to make safer driving decisions and navigate roads more effectively. This helps to reduce the risk of accidents and improve overall road safety.

Secondly, dynamic maps provide detailed information about the road layout, traffic lights, road signs, and other key features that can impact the vehicle’s path. This information is crucial for enabling autonomous vehicles to make more informed navigation decisions, reducing the risk of missed turns, missed traffic signals, and other navigation errors. This improved navigation capability is essential for ensuring the safe and efficient operation of autonomous vehicles [6].

Thirdly, dynamic maps enhance the situational awareness of autonomous vehicles, allowing them to have a better understanding of their surroundings. This improved awareness helps to reduce the risk of accidents and improve overall road safety by enabling autonomous vehicles to respond more effectively to changes in the driving environment.

Fourthly, dynamic maps are constantly updated in real time, providing the most recent information to autonomous vehicles. This allows autonomous vehicles to react to changing conditions and make more informed decisions, ensuring that they are able to respond to real-time changes in road conditions, traffic patterns, and other important factors. This real-time updating capability is crucial for ensuring the effective operation of autonomous vehicles in dynamic and unpredictable driving environments [7].

Finally, the use of dynamic maps helps to ensure the reliability of autonomous vehicle technology. By providing accurate, up-to-date information, dynamic maps help to minimize the risk of navigation errors and other operational issues, which can negatively impact the performance of autonomous vehicles. The reliability of autonomous vehicle technology is a critical factor in ensuring the safe and effective deployment of autonomous vehicles [8].

In conclusion, dynamic maps play a crucial role in the development and deployment of autonomous vehicles. By providing real-time information about the driving environment, dynamic maps help to ensure the safety, reliability, and effectiveness of autonomous vehicle technology. The integration of dynamic maps into autonomous vehicles is essential for ensuring the safe and efficient operation of these vehicles in real-world driving environments [9].

Figure 1 provides an illustration of the utilization of GPS-based waypoint methodologies in the creation of dynamic maps for autonomous vehicles, which has historically been the prevalent technique. However, this method has limitations in accurately representing the road environment and recognizing objects such as road signs, lane markings, and traffic signals, which are crucial for the vehicle’s navigation and decision making. The quality and accuracy of GPS signals can be impacted by interference from buildings, trees, and other objects, leading to inaccuracies in the vehicle’s understanding of its surroundings and negatively impacting safety and the overall performance of autonomous driving systems [10].

To overcome the limitations inherent in GPS-based waypoint methods, the research and development community has been investigating alternative mapping technologies that incorporate a fusion of various sensors such as cameras, LIDAR, and RADAR to create a more comprehensive and precise representation of the road environment. These high-definition maps can be dynamically updated in real time to incorporate changes in the driving conditions, making them a crucial component for the secure and successful deployment of autonomous vehicles [11].

In this context, this paper proposes a method for constructing a dynamic high-definition map for autonomous driving with the objective of enhancing accuracy and safety. The proposed method employs the use of multiple sensors to gather autonomous driving route data, which are then processed to generate a dynamic high-definition map. The generated map is supplied to the autonomous vehicle, enabling it to navigate through the designated route with greater precision and confidence.

The integration of multiple sensors in the vehicle, in conjunction with prior knowledge of the autonomous driving route, allows the technology to continuously update the map with the minimum necessary changes, thereby mitigating real-time driving challenges. This dynamic map serves as a critical resource for autonomous vehicles, providing them with a constantly updated representation of the driving environment [12].

It is hypothesized that the provision of this dynamic high-definition map to autonomous vehicles will result in significant improvements in both the reliability and safety of autonomous driving technology. This hypothesis is based on the premise that a more comprehensive and up-to-date representation of the driving environment will enhance the autonomous vehicle’s ability to navigate safely and efficiently.

## 2. Mobile Mapping System (MMS)

Figure 2 presents the multi-sensor-based mobile mapping system (MMS) utilized to acquire autonomous driving routes. The MMS comprises of six high-resolution cameras, a global navigation satellite system (GNSS), and high-resolution LIDAR sensors [13]. The six cameras equipped in the MMS gather comprehensive information such as the textures and hues of the surrounding environment and are capable of capturing over 80% of a 360-degree panoramic view with a shooting rate of 30 frames per second (FPS) or higher and a resolution of 2448 × 2048 or higher. Furthermore, each of the six cameras provides a resolution of greater than five million pixels [14].

The GNSS integrated within the MMS furnishes data regarding the location, velocity, and orientation of the space being captured and guarantees a high-precision performance of 0.03 m by ensuring a tight match between the data provided and the actual location.

The high-resolution LIDAR device integrated within the MMS has a single channel with a capability of 1,000,000 points per second, a maximum measurement range of 120 m, and records approximately 1900 points per 1 square meter of area when the vehicle is in motion at a speed of 30 km/h [15].

## 3. Data Acquisition and Processing Method with the MMS

Figure 3 shows the vehicle equipped with the MMS when it is driven. Acquisition scenarios or procedures are established to acquire spatial information using the multi-sensor-based MMS [16]. Acquisition scenarios and planning are needed to obtain various types of data from the different sensors and to display accurate locations and spaces with point clouds or image-processing programs through data synchronization and merge operations.

Figure 4 shows a flow chart of the multi-sensor-based MMS data-acquisition system. This paper describes a method for constructing a dynamic map and multi-sensor-based mobile mapping system (MMS) [17]. The methodology is depicted in a flowchart as presented in Figure 4. The process consists of three distinct phases: (1) planning, in which the route for collecting the data required for the dynamic map construction is devised, ensuring that it can be traversed in a single pass manner; (2) data acquisition, where the vehicle equipped, such as in Figure 3, with the MMS equipment travels along the planned route and collects the required data, and to ensure the accuracy of the collected data, the sensors are synchronized prior to data acquisition, and additional data are collected at the start and end points to facilitate sensor synchronization; (3) data processing, where the acquired data are processed, edited, and adjusted to a specific size, and if there is insufficient inter-point spacing between the start and end points of data acquisition, it may result in data degradation and the subsequent inability to construct an accurate dynamic map; and (4) converting data, where the acquisition dataset is converted into the LAS file format. The acquisition dataset is converted into processed data (the LAS file format) that can be plotted on a precision map through synchronization with a standard GNSS base station for the objectification of the acquisition data by fusing the raw data from the LIDAR sensor and the camera with the GNSS’s moving-path raw data.

Figure 5 illustrates the phenomenon of overlapping, which occurs when the route is traversed more than twice during the planning stage, as depicted in Figure 4. The overlapped data are depicted as overlapped traffic sign markers. Vehicles equipped with the multi-sensor-based MMS cannot travel in the same areas under the same conditions (e.g., driving speed and road position between driving vehicles). Therefore, differences in the data obtained from the LIDAR sensor installed in the MMS may occur, resulting in differences and errors, as shown in Figure 5. However, when drawing a dynamic high-definition map, if some of the overlaid data are selected and drawn consistently, there will be no problems when drawing a dynamic high-definition map. In order to enhance the consistency and accuracy of precision map data, it is crucial to acquire the data by traversing the planned route in a single pass.

Figure 6 represents the raw GNSS-based movement path data obtained in the acquisition of spatial data. The illustration represents the space in which data were collected utilizing a vehicle equipped with the multi-sensor mapping system (MMS), with the aim of creating a high-definition dynamic map. Upon conversion into processed data in the LAS file format, the raw movement path data retain their original form.

Figure 7 depicts the synchronization of GNSS observation values, which serve as the standard, with the raw data acquired from the GNSS-based movement path. In Figure 8, the acquired data are synchronized through the utilization of ten or more standard GNSS observation values in proximity to the official observation. The integration of the official observation values, represented by green points, with the movement path of the raw data, represented by purple lines, leads to an improvement in the accuracy of the GNSS data. The result of this synchronization process is the conversion of the raw data of the GNSS-based spatial acquisition into vector data, which are characterized by their spatial coordinate data components along the *x*-, *y*-, and *z*-axes, thus transforming the data from a general three-dimensional representation.

Figure 8 presents the integration of the vector-processed data, the raw data obtained from LIDAR measurements, and the raw data obtained from the camera, all with their respective spatial coordinate components. The position parameter value of the LIDAR data is adjusted based on the height and length of the vehicle equipped with the MMS equipment to ensure accuracy. The result of this correction is the combination and storage of over one million LIDAR data points in the LAS file format as processed data. The raw data from the camera are also transformed into a panoramic image in the JPG format, with a resolution of 2448 × 2048 pixels.

Figure 9 depicts the representation of the processed data in the form of LAS files, which serve as the fundamental data in the mapping domain, with spatial coordinate components in the *x*-, *y*-, and *z*-axes. Unlike conventional two-dimensional online maps, such as Google Maps, these processed data form a three-dimensional spatial map representation, providing a comprehensive and tangible spatial representation. The LAS file-format-based machined data are crucial in navigation or in the development of dynamic high-definition maps for autonomous driving, as they integrate information from LIDAR, camera, and GNSS data.

Finally, the production of a dynamic high-definition map is based on the data processed in the LAS file format. The LAS file format, as it encompasses the spatial coordinate data components of the *x*-, *y*-, and *z*-axes, yields a highly precise map that includes both directional and positional information crucial for autonomous navigation. As a result, the machined data in the LAS file format serve as the foundation for the creation of a base map for autonomous vehicles.

## 4. Dynamic High-Definition Map-Drawing Method

Figure 10 illustrates the process of constructing a dynamic high-definition map. The data processed in the LAS file format are represented in a three-dimensional space that includes road lanes and traffic signs. This representation is essential to ensure the effective utilization of the *z*-axis coordinate, which provides the height information, in navigation and autonomous driving applications. Without a three-dimensional representation, the dynamic high-definition map would not be able to incorporate the *z*-axis data, leading to reduced accuracy and stability in navigation and autonomous driving systems. Therefore, it is necessary to utilize a viewer or software capable of visualizing data in three dimensions when creating a dynamic high-definition map.

Figure 11, Figure 12 and Figure 13 present the results of dynamic high-definition mapping in various regions. The dynamic high-definition maps are depicted as spatial vector lines, with each line incorporating information on movement direction, precise location, and height as attributes. The lines and planes displayed in the dynamic high-definition maps are defined in accordance with the needs of autonomous driving technology or parsing programs.

The dynamic high-definition map for autonomous driving, as proposed in this technology, is capable of providing structured information regarding the road network and its environment, including fixed features such as lane boundaries and traffic signs, as well as data about objects that change in real time, such as other vehicles or obstacles. This structured information is obtained through the fusion of data from multiple sources such as LIDAR, cameras, and GNSS, and it is represented in a three-dimensional space, which allows for better accuracy and stability. By limiting the amount of real-time information and prioritizing fixed data, the proposed technology reduces the computational and data processing requirements, leading to improved performance and reliability of autonomous driving systems.

## 5. Result

Figure 14 represents the process of incorporating high-precision map data into an autonomous vehicle. The proposed system outlines the autonomous driving route based on the high-precision map data. Unlike traditional autonomous driving systems, there is no need for multiple sub-module sensor systems or coordinate data sheets. This is because the high-precision map data produced by the proposed system already contain GNSS and surrounding object data. In contrast, additional data could actually hinder the autonomous driving process by increasing processing time. The various sensors installed in the autonomous vehicle are only necessary for recognizing changes in the traffic environment and objects along the high-precision map and autonomous driving route in real time.

Figure 15 represents a proposed autonomous vehicle system that leverages a precision road map constructed by the system. In contrast to the conventional waypoint-based precision road map (as shown in Figure 1), the proposed system has the capability of perceiving its environment in three dimensions, thereby enabling real-time representation of the surroundings on the precision road map. This enhances the system’s ability to respond to the dynamic traffic conditions and unforeseen scenarios. The conventional waypoint-based precision road map, however, is limited to a two-dimensional representation of only waypoints, which hinders its real-time responsiveness and ability to address unexpected situations. Moreover, the existing system necessitates the use of a separate interface to verify the autonomous vehicle’s recognition of surrounding objects, which constitutes a significant inconvenience. In contrast, the proposed system benefits from its ability to recognize objects in real time through the digital representation of the physical space in three dimensions.

Therefore, the proposed system enhances the real-time perception of autonomous driving by providing data and information that allow the system to quickly perceive and respond to changing traffic conditions and unexpected situations. The precision road map based on the proposed system is superior to the existing waypoint-based precision road map in terms of real-time object recognition and traffic environment recognition performance, thereby further improving the reliability and safety of autonomous driving. As a result, the utilization of the proposed system results in the advancement of autonomous driving capabilities.

Figure 16 represents the autonomous vehicle that is navigating using a precision map constructed using the proposed system. Utilizing the MMS equipment, the precision map data of the intended route were constructed and fed into the autonomous vehicle. The sensors installed on the autonomous vehicle were employed to recognize the surrounding objects and traffic environment and respond in real time by reflecting the situation in the input precision map. In Figure 17, while the autonomous vehicle is navigating, other vehicles are detected on different paths, and the current driving path only contains roads, with no external objects such as traffic signs or pedestrians present, allowing the vehicle to continue navigating smoothly.

Figure 17 represents a comparison between the conventional system and the proposed system. The conventional system is a waypoint-based system utilizing GPS, which requires manual implementation of all functions prior to the start of autonomous driving. Firstly, the waypoints must be accurately entered into the autonomous driving route, and it must be verified through the user interface by pressing buttons on the sensor control panel to confirm that the sensors installed on the autonomous vehicle are functioning properly. Additionally, there is an inconvenience of having to verify through the vehicle control panel that the vehicle is communicating accurately with the existing system.

However, the proposal system inputs a high-precision road map on which autonomous driving routes are generated. The autonomous vehicle then recognizes surrounding objects and traffic conditions and displays them on the precise road map. Based on the recognized sensor data, the autonomous driving information and routes are generated and transmitted to the autonomous vehicle. As a result, the autonomous vehicle executes autonomous driving, and the sensors installed in the autonomous vehicle continuously recognize the external environment to enhance the performance and reliability of autonomous driving.

The proposed system has the advantage over the existing system in real-time recognition of external environmental objects and traffic environment during autonomous driving. Furthermore, it is able to quickly respond to sudden situations that may occur in the autonomous driving environment. Additionally, the proposed system has the advantage of providing passengers with a more intuitive and accurate depiction of the autonomous vehicle’s state, direction of movement, and route compared to the existing system, making it easier for them to understand. As a result, the proposed system is capable of offering a safer and more reliable autonomous driving environment than the conventional system.

## 6. Conclusions

This paper presents a novel method for constructing a dynamic high-definition map for autonomous driving using a multi-sensor-based mapping system (MMS). The proposed method leverages multiple sensors to acquire data on the autonomous driving route, which are then processed to generate an accurate and reliable dynamic high-definition map. The results of this study demonstrate that the dynamic high-definition map produced by the proposed method offers a significant advantage over conventional systems in terms of real-time object recognition and traffic situation analysis. With an error range of 0.03 m, the proposed method offers a higher level of accuracy compared to the standard error range of 0.25 m.

The use of a dynamic high-definition map in autonomous driving systems provides a more intuitive and accurate representation of the driving environment, enabling passengers to have a better understanding of the vehicle’s status, direction, and path. This enhances the overall safety and reliability of the autonomous driving experience.

In terms of future research, there is a need to further improve the precision of the dynamic high-definition map and to develop new algorithms for robust and efficient object recognition and traffic situation analysis. Additionally, the integration of this system with advanced technologies such as machine learning and big data analysis has the potential to enhance the overall performance of autonomous vehicles. These efforts will play a crucial role in advancing the field of autonomous driving and ensuring a safer and more reliable future for this technology.

## Figures and Tables

**Figure 1 sensors-23-02369-f001:**
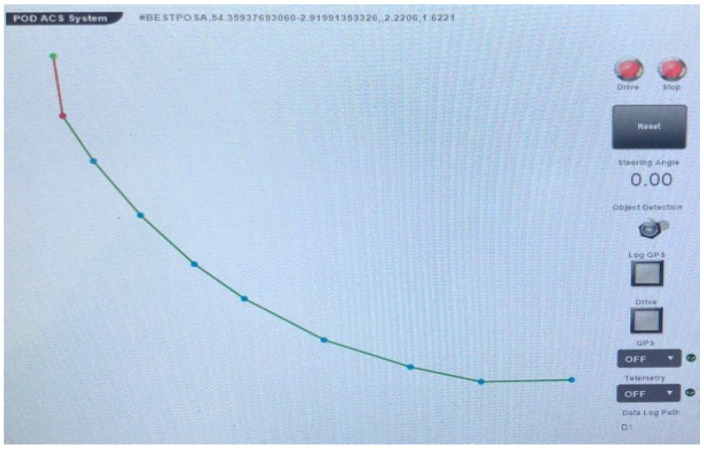
Example of a conventional dynamic map based on GPS waypoint method.

**Figure 2 sensors-23-02369-f002:**
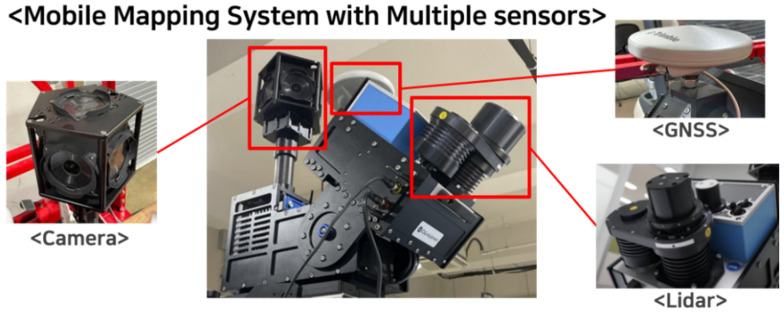
Mobile mapping system (MMS) with multiple sensors.

**Figure 3 sensors-23-02369-f003:**
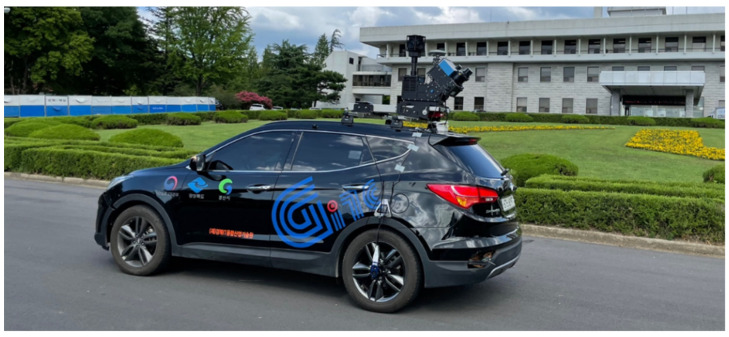
Driving the vehicle with the MMS.

**Figure 4 sensors-23-02369-f004:**
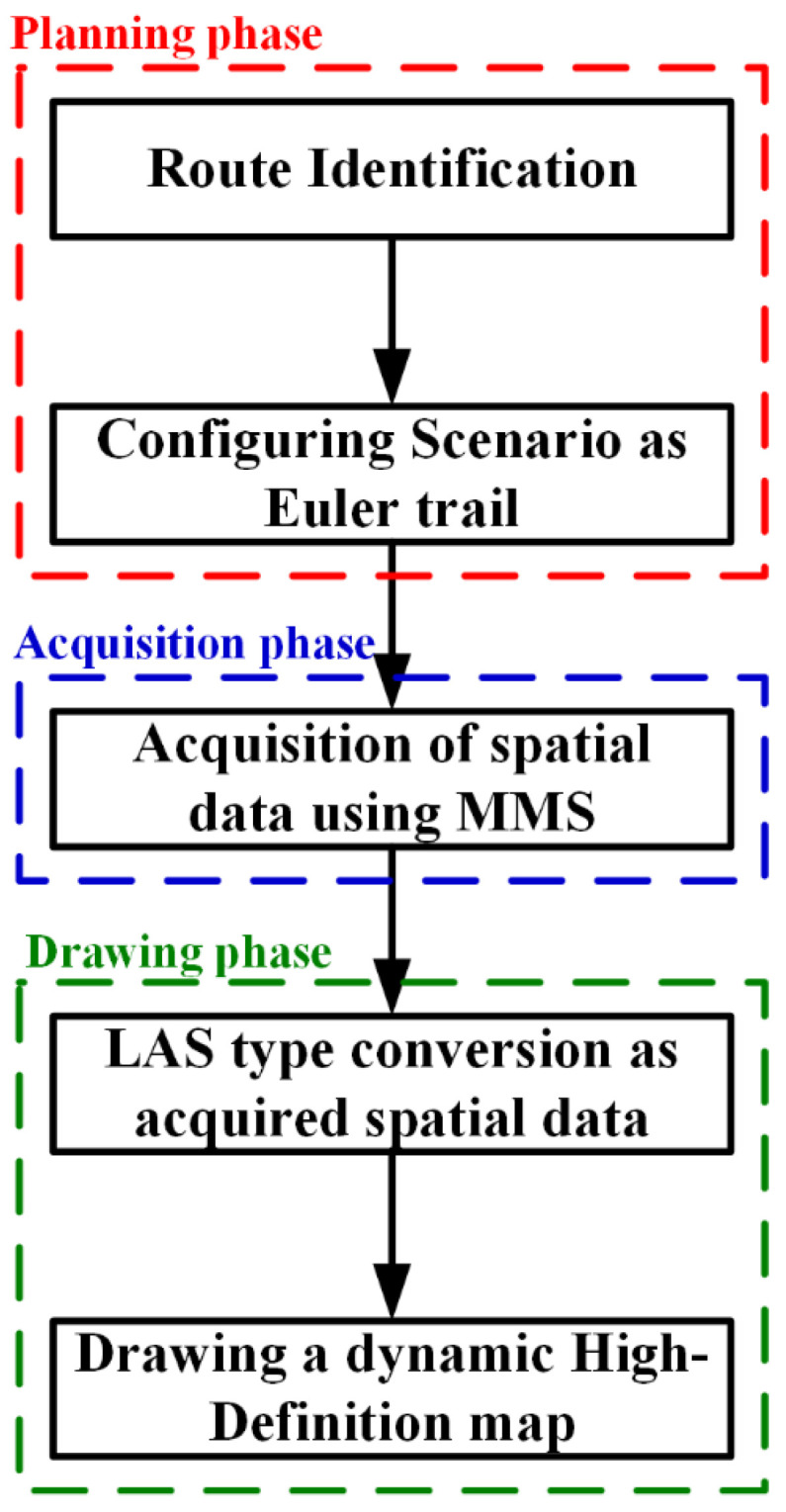
Flow chart of acquisition data using the MMS.

**Figure 5 sensors-23-02369-f005:**
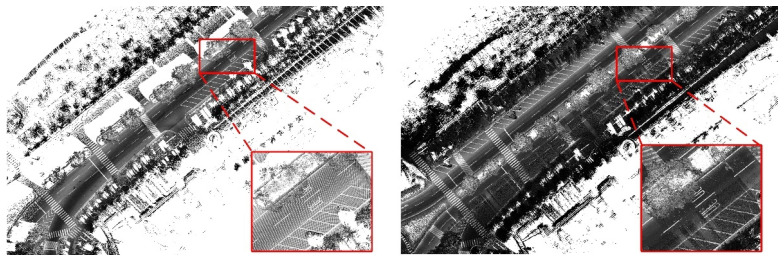
Superposition data in the point cloud.

**Figure 6 sensors-23-02369-f006:**
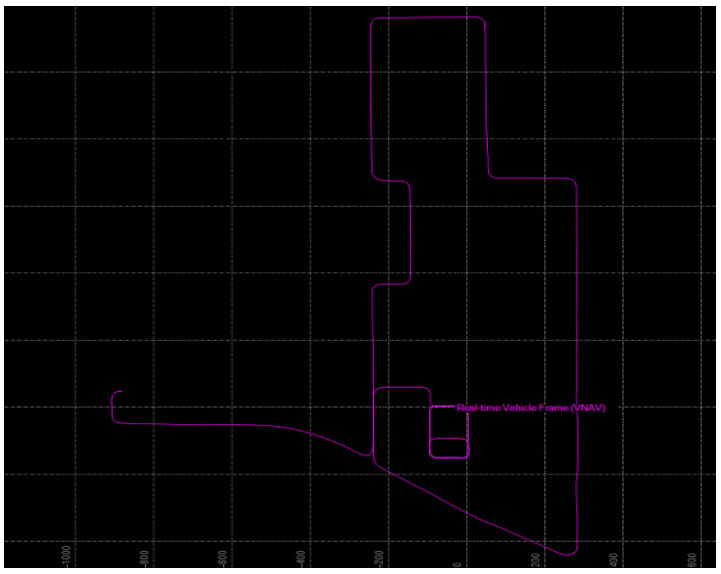
Example of raw data for the GNSS movement path.

**Figure 7 sensors-23-02369-f007:**
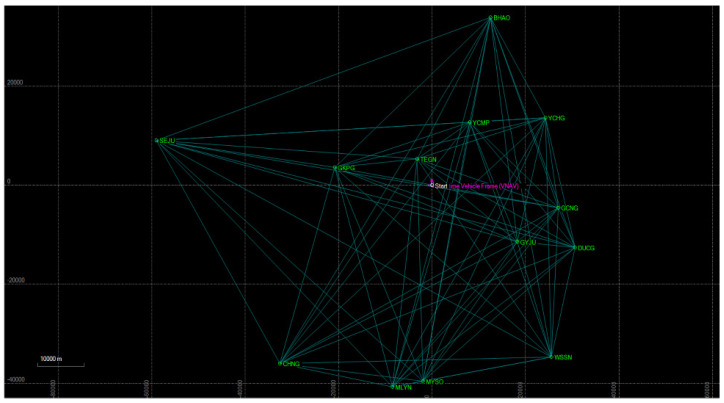
Example of synchronizing the GNSS movement path raw data with standard GNSS observations.

**Figure 8 sensors-23-02369-f008:**
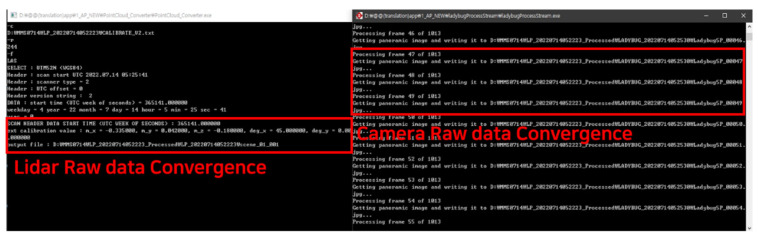
Example of the convergence of LIDAR raw data and camera raw data.

**Figure 9 sensors-23-02369-f009:**
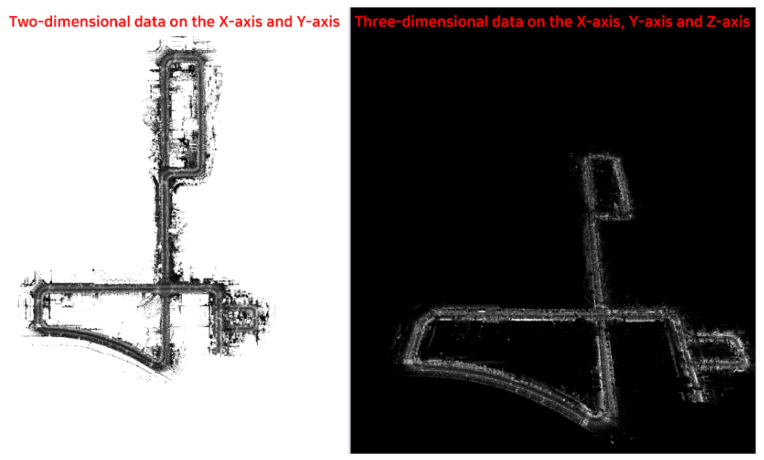
Example of machining data based on the LAS file format.

**Figure 10 sensors-23-02369-f010:**
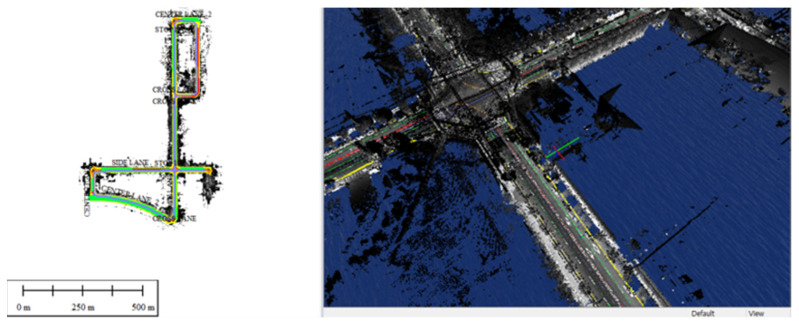
Dynamic high-definition map-drawing process.

**Figure 11 sensors-23-02369-f011:**
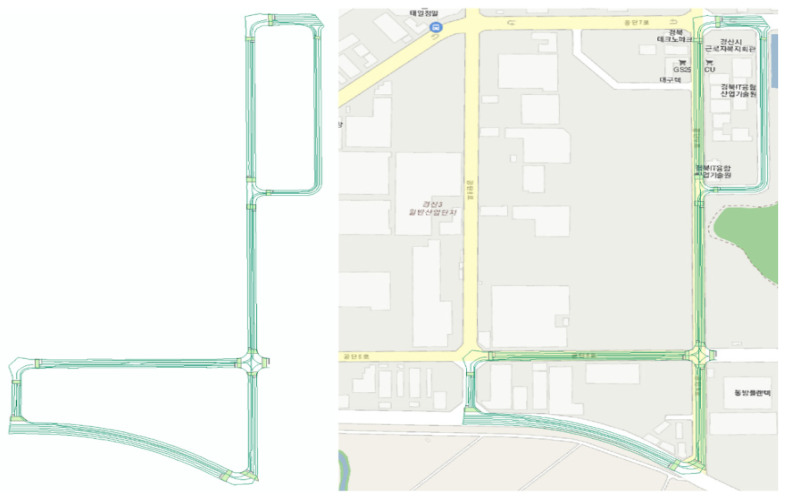
Dynamic high-definition map result in an industrial park.

**Figure 12 sensors-23-02369-f012:**
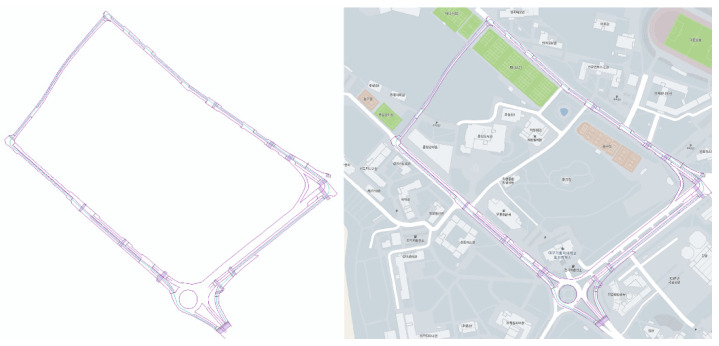
Dynamic high-definition map result at a university.

**Figure 13 sensors-23-02369-f013:**
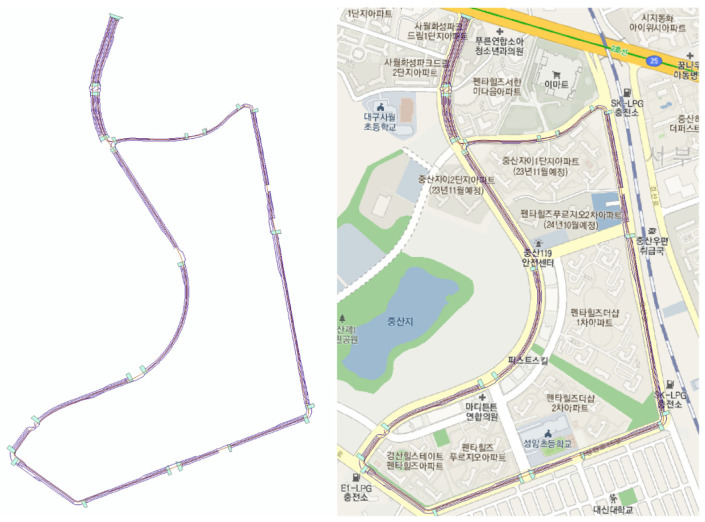
Dynamic high-definition map result at a residential complex.

**Figure 14 sensors-23-02369-f014:**
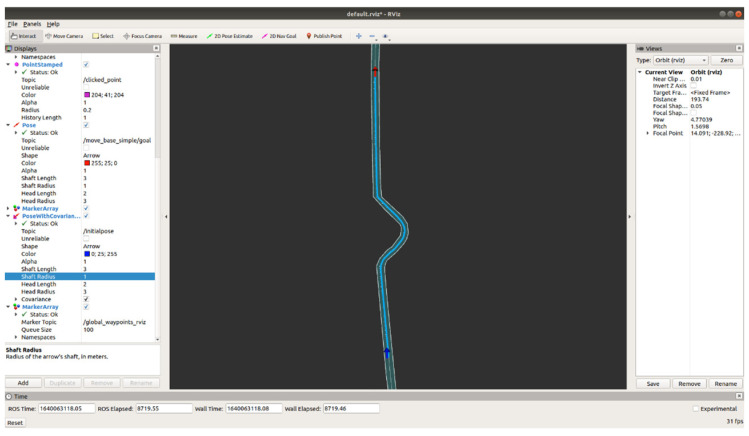
The process of incorporating high dynamic map data into an autonomous vehicle.

**Figure 15 sensors-23-02369-f015:**
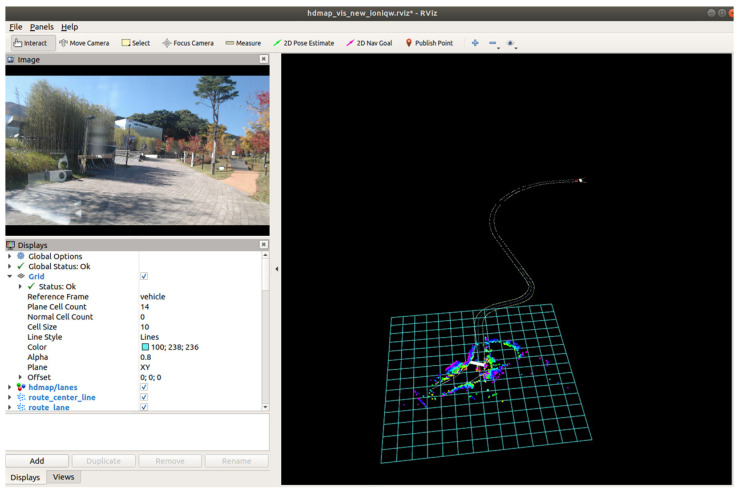
Autonomous Driving System Based on a Precision Road Map Constructed by the Proposed System.

**Figure 16 sensors-23-02369-f016:**
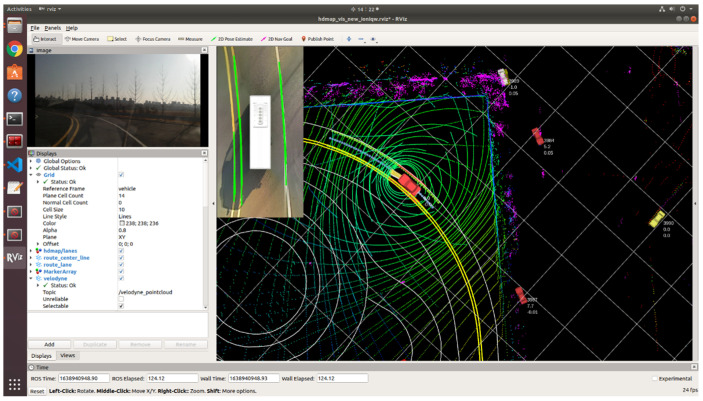
Application of a Proposed Precision Geospatial Mapping System for Autonomous Driving: A Case Study.

**Figure 17 sensors-23-02369-f017:**
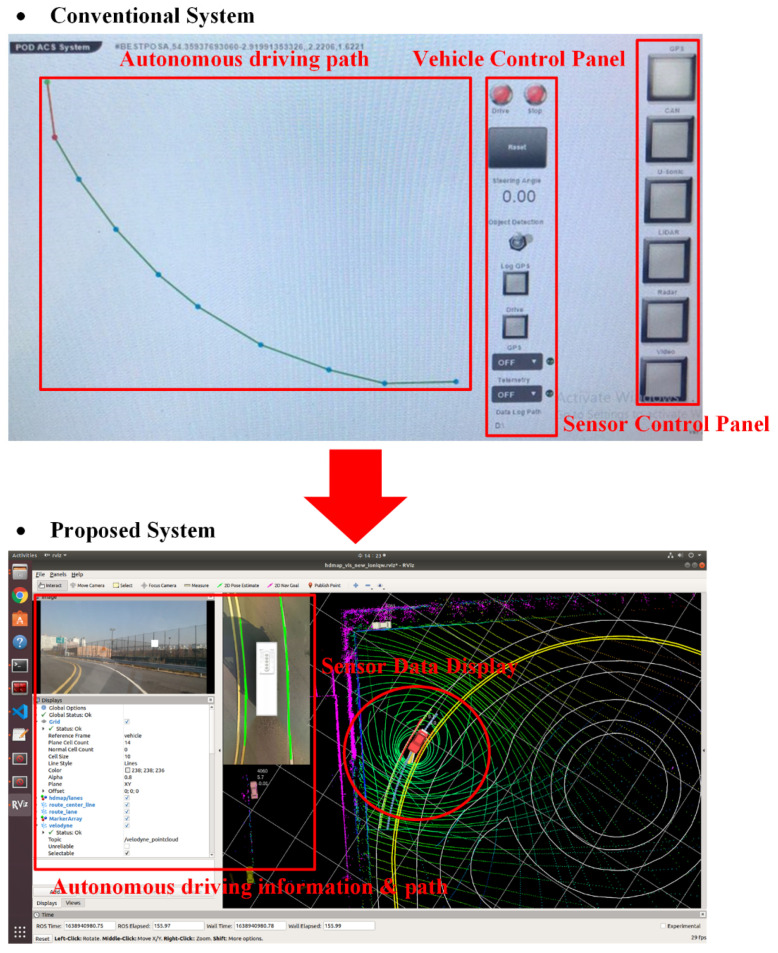
A Comparison of Conventional and Proposed Autonomous Driving Systems in Implementation.

## Data Availability

Publicly accessible repository.

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
