# Peer review of "Investigating the Improvement of Autonomous Vehicle Performance through the Integration of Multi-Sensor Dynamic Mapping Techniques"

_sensors, 2023, doi:10.3390/s23052369_

Round 1

Reviewer 1 Report (Previous Reviewer 3)

The content of this paper is about the research on the performance improvement of autonomous vehicle based on multi-sensor dynamic mapping technology. The proposed method uses dynamic high-definition maps to improve the recognition rate of objects near the vehicle and automatic driving path recognition, and uses multiple sensors, such as cameras, laser radars and radars. The research content has important practical significance. The overall content is logical. There are still some deficiencies in the format and content that need to be modified.

1.The background introduction in the introduction is too rigid and the cited documents are not smooth.

2.Whether line 64 can be rewritten into a dynamic map.

3.The contents of Figure 7 and Figure 8 are seriously blurred and can not be seen clearly.

I suggest that this paper be accepted after revision.

Author Response

Reviewer 2 Report (New Reviewer)

This manuscript proposes a multi-sensor dynamic mapping algorithm. It is interesting and meaningful. This work is an indispensable part of the implementation of self-driving cars. Some comments are given below:

1: I think Figure 1 is not your own figure. If so, you should add the corresponding reference.

2: The introduction is too simple and should be improved. For mobile mapping systems with multiple sensors, time synchronization, and lever arm calibration are challenging for performance improvement. please consider these highly related works to optimize the introduction: Automated vehicle sideslip angle estimation considering signal measurement characteristic; IMU-based automated vehicle body sideslip angle and attitude estimation aided by GNSS using parallel adaptive Kalman filters.

3. Through the multi-sensor dynamic techniques, it would provide the vehicle with the surrounding environment information and ego state information. Especially, the ego state information especially the velocity and attitude are essential for vehicle dynamic control to ensure stability. But you ignored this important work. Some work could be found in: Intelligent vehicle sideslip angle estimation considering measurement signals delay; Visionaided intelligent vehicle sideslip angle estimation based on a dynamic model. It would be meaningful to add the corresponding analysis to the paper. In addition, would you mind explaining why you don’t use the sensor of IMU?

4. What are the limitations of your work?

5. Please polish the figures. Note the uniform font size and background color.

6. Please define the coordinate systems of different submodule-sensor systems on the vehicle in advance. In addition, List the datasheet of different sensors such as accuracy and bias. In this way, readers easily understand the sensor performance information of the autonomous driving platform.

Author Response

This manuscript is a resubmission of an earlier submission. The following is a list of the peer review reports and author responses from that submission.

Round 1

Reviewer 1 Report

In this article, a multi-sensor fusion method is used to construct a dynamic high-definition map for autonomous driving to improve the accuracy and stability of autonomous driving technology. However, the lack of necessary data support and experimental comparison with other methods in the article makes it difficult to prove that the accuracy of autonomous driving technology has been improved.

Reviewer 2 Report

In this paper, the authors tackled an important and timely problem of generating high-definition maps for autonomous traffic. There is no question that the topic is worthy of investigation, but the manuscript suffers from the following issues which, in my opinion, should be thoroughly tackled before it could be considered for publication:

1.      The manuscript is poorly written and should be thoroughly proofread with a help of a native speaking colleague, as there are lots of sentences which are challenging to understand (also in the abstract).

2.      Please avoid having paragraphs which contain a single sentence only.

3.      Were the figures prepared by the authors or are there taken from other (Internet/papers) sources? It looks like it is the latter case – the authors should not use the figures taken directly from other sources.

4.      The quality of figures should be improved – they should be in a vector format.

5.      The manuscript lacks novelty and contextualization within the state of the art – there are methods for building the maps of surrounding and high-definition maps in (not only) autonomous vehicles.

6.      The paper presents a very high-level description of the system (it may rather be considered a teaser), not a detailed design of the approach.

7.      The number of captured real-life examples should be presented in the manuscript.

Reviewer 3 Report

This paper proposes a method for constructing dynamic high-definition maps for multi-sensor autonomous driving that can improve the accuracy of autonomous driving, with a relatively new intention. However, this cannot cover up the author's extremely perfunctory attitude towards science. The lack of innovation or incorrect experimental methods in the paper is the author's scientific ability that needs to be improved, and the paper has so many problems that it even affects the reading, reflecting the author's extremely improper scientific attitude.
1. Figure 1 and Figure 2 color scheme is not reasonable, the word is small and unclear, the advantages and disadvantages of Figure 2 is not expressed clearly.
2. Can the arrows in Figure 3 be matched with the words? This looks extremely perfunctory and unintentional; can the part photo in Figure 4 seem less comprehensive than the system photo, can it be expressed more clearly? The path lines in Figure 8 and Figure 9 are not even as clear as the grid lines in the background.
3. It is mentioned in line 83 that there are still many problems hindering autonomous driving technology, what are they? Should they be described in detail? Is there any basis for the previous discussion? Where are they reflected?
4. The flow chart of MMS data collection shown in Figure 6 should be described in detail.
5. Why some diagrams are in the front and explanation is in the back, and some explanation is in the front and diagram is in the back, which looks disorganized, can the form be unified?
6. This article has too few references, and some arguments are not even based, how to prove the reliability and accuracy?

I strongly recommend rejecting this article. The author of this article has a very poor scientific attitude, the method is not clearly introduced and the proof is not rigorous, which seriously affects the accuracy of the final conclusion.